# Aerosol Delivery of Dornase Alfa Generated by Jet and Mesh Nebulizers

**DOI:** 10.3390/pharmaceutics12080721

**Published:** 2020-07-31

**Authors:** Kyung Hwa Chang, Sang-Hyub Moon, Sun Kook Yoo, Bong Joo Park, Ki Chang Nam

**Affiliations:** 1Department of Medical Engineering, Dongguk University College of Medicine, Goyang-si, Gyeonggi-do 10326, Korea; chang9429@hotmail.com (K.H.C.); shnoom@naver.com (S.-H.M.); 2Department of Medical Engineering, Yonsei University College of Medicine, Seoul 03722, Korea; sunkyoo@yuhs.ac; 3Department of Electrical & Biological Physics, Kwangwoon University, Seoul 01897, Korea; 4Institute of Biomaterials, Kwangwoon University, Seoul 01897, Korea

**Keywords:** nebulizer, dornase alfa, Pulmozyme, aerosol size, output rate, DNase I activity

## Abstract

Recent reports on mesh nebulizers suggest the possibility of stable nebulization of various therapeutic protein drugs. In this study, the in vitro performance and drug stability of jet and mesh nebulizers were examined for dornase alfa and compared with respect to their lung delivery efficiency in BALB/c mice. We compared four nebulizers: two jet nebulizers (PARI BOY SX with red and blue nozzles), a static mesh nebulizer (NE-U150), and a vibrating mesh nebulizer (NE-SM1). The enzymatic activity of dornase alfa was assessed using a kinetic fluorometric DNase activity assay. Both jet nebulizers had large residual volumes between 24% and 27%, while the volume of the NE-SM1 nebulizer was less than 2%. Evaluation of dornase alfa aerosols produced by the four nebulizers showed no overall loss of enzymatic activity or protein content and no increase in aggregation or degradation. The amount of dornase alfa delivered to the lungs was highest for the PARI BOY SX-red jet nebulizer. This result confirmed that aerosol droplet size is an important factor in determining the efficiency of dornase alfa delivery to the lungs. Further clinical studies and analysis are required before any conclusions can be drawn regarding the clinical safety and efficacy of these nebulizers.

## 1. Introduction

Dornase alfa (proprietary name: Pulmozyme, from Genentech) is a highly purified solution of recombinant human pancreatic deoxyribonuclease I (rhDNase I), an enzyme that selectively cleaves DNA. It was found to improve the rheological properties of purulent cystic sputum from cystic fibrosis (CF) patients in vitro [1]. The reduction of viscosity is due to a decrease in the length of DNA derived from human neutrophils, which infiltrate the airways to eliminate pathogenic bacteria. The long-term effects of dornase alfa on lung exacerbation and its short-term (not more than 14 d) and long-term effects on lung function have been studied in clinical trials. These trials showed that dornase alfa is well tolerated, significantly improves lung function, and reduces the risk of pulmonary exacerbations [2,3,4,5,6,7,8,9]. Thus, dornase alfa is currently used as a mucolytic agent to treat the leading cause of morbidity and mortality in CF. According to the guidelines issued for chronic respiratory medication in CF patients, chronic patients aged over 6 years should inhale a 2.5 mg single-use ampule once daily using a recommended nebulizer [10].

Nebulizers, which are mainly used for aerosol therapy, can agitate liquid medication to create small aerosol droplets that can rapidly deliver drugs directly to the lungs and thus enable local drug delivery for the topical treatment of respiratory diseases, reducing the risk of systemic side effects [11]. In addition, nebulizers do not require coordination between inhalation and actuation; thus, they are useful for pediatric, elderly, ventilated, and non-conscious patients [12]. There are three types of nebulizers—jet, ultrasonic, and mesh nebulizers—which differ in terms of their working principles. Jet nebulizers are generally considered the gold standard method in the clinical field for delivering medicines to patients requiring aerosol therapy [13,14]. Current product information on dornase alfa for aerosol therapy in CF patients only recommends six jet nebulizers and one vibrating mesh nebulizer [15]. This is because jet nebulizers have been used to administer dornase alfa in all clinical studies except for one, in which the Pari eRapid system (a portable vibrating mesh nebulizer) was evaluated [16].

Jet and ultrasonic nebulizers have been developed as the primary type, but ultrasonic nebulizers are more expensive than jet nebulizers and tend to increase the temperature of the nebulized drug solution. Thus, they are considered inappropriate for the nebulization of thermolabile peptides and proteins. On the other hand, compressor-based jet nebulizer systems are inconvenient for patients because they require extra tubing, a heavy compressor, and long treatment times, and are also noisy to use and relatively inefficient due to their large residual volumes [12,13,14]. Compared with jet and ultrasonic nebulizers, mesh nebulizers are more portable, far more convenient for patients, deliver all liquid in reservoirs (i.e., zero residual volume), and enable efficient, rapid, and reproducible delivery of small drug volumes to the lungs. In a clinical trial, the eRapid mesh nebulizer system was strongly preferred by patients over the standard jet nebulizer [16].

Mesh nebulizers can be divided into two types: static (passive) and vibrating (active) nebulizers. Static mesh nebulizers use a horn that vibrates ultrasonically against a static mesh, while vibrating mesh nebulizers use a mesh mounted in an ultrasonically vibrating piezo ring [13,17,18]. Recently, the performance characteristics of mesh nebulizers have attracted considerable interest for the pulmonary delivery of expensive protein-based biopharmaceuticals. Several studies have reported successful nebulization of sensitive drugs, such as proteins and antibodies, by mesh nebulizers. These in vitro studies, which were performed using a breathing simulator, showed that the delivery efficiency of protein aerosols produced by mesh nebulizers is comparable to that of jet mesh nebulizers [16,19,20,21,22,23,24]. However, they did not compare the dose delivered to the lungs using an in vivo model. Although a variety of mesh nebulizers are commercially available, only the eRapid nebulizer has been approved for use with dornase alfa.

Mesh nebulizers utilize the same piezoelectric transducer used in ultrasonic nebulizers to create aerosol droplets. In static nebulizers, this piezo element is mounted in a horn, whereas in vibrating mesh nebulizers, it is combined with the mesh substrate [25]. It has been reported that piezoelectric elements in ultrasound nebulizers can produce sufficient heat to cause protein denaturation and loss of activity [26,27,28,29]. Recent reports show that the temperature of the liquid in the reservoir of a mesh nebulizer increases during nebulization [30,31]. Therefore, detailed studies are needed to determine whether heat generated by mesh nebulizers during operation affects protein stability.

This study was undertaken to investigate the in vitro performance of four commonly used home nebulizers—two jet nebulizers (PARI BOY SX with red and blue nozzles), a static mesh nebulizer (NE-U150), and a vibrating mesh nebulizer (NE-SM1)—and to compare their effects on enzyme activity, structural integrity, temperature change, and in vivo delivery efficiency, using dornase alfa as the test drug. Dornase alfa delivery efficiency was evaluated using spontaneously breathing mice without mechanical ventilation or breathing masks in a closed inhalation chamber.

## 2. Materials and Methods

### 2.1. Materials and Nebulizers

Commercially available nebulizer solutions of 2.5 mg/2.5 mL dornase alfa were acquired from Genentech, Inc (South San Francisco, CA, USA) by Material Transfer Agreements (ID: OR-215976). The DNaseAlert™ QC System was obtained from Thermo Fisher Scientific (Waltham, MA, USA). The DC protein assay kit and 4–20% Mini-PROTEAN^®^ TGX™ Precast Protein Gels were obtained from Bio-Rad Laboratories (Hercules, CA, USA). Anti-DNase I and horseradish peroxidase (HRP)-conjugated goat anti-rabbit IgG were purchased from Abcam (Cambridge, MA, USA) and Cell Signaling Technology (Danvers, MA, USA), respectively

Four home nebulizers were investigated; the operating modes and abbreviations used for each nebulizer are described in Table 1. Since the recommended devices for dornase alfa were not approved in the Korean market, popular items were selected for this study among the nebulizers commonly used in the clinic. PARI BOY SX in this study is the closest model to PARI LC PLUS, which is one of the recommended devices for dornase alfa. The PARI BOY SX jet nebulizer was used with two types of nozzle (red and blue); both were supplied by the manufacturer. According to the manufacturer, the red nozzle attachment creates finer droplets than the blue nozzle.

### 2.2. Animals

All animal experiments were conducted in accordance with guidelines issued by the Ethics Committee of Animal Service Center at Dongguk University (IACUC Number: 2019-010-1, 18 April 2019). Female BALB/c mice aged seven to eight weeks were obtained from Daehan Biolink (Eumseong, Korea). The mice were housed in standard cages and maintained under controlled, constant room temperature (RT) and humidity with a 12-h light/dark cycle. The animals had free access to both water and food.

### 2.3. Size of Nebulized Aerosol Droplets

The aerosol size distributions were assessed using a Spraytec (Model #STP5311, Malvern instrument, Malvern, UK), which utilizes the laser diffraction method and has been demonstrated to be a reliable and time-saving method [32]. In each experiment, saline and 2.5 mL aliquots of 1 mg/mL dornase alfa were placed in nebulizers. In order to measure aerosol droplet sizes, all nebulizers were placed into a Spraytec without mask adapters or mouthpieces. Aerosol droplet sizes were measured during nebulization for 1 min after an initial nebulization period of 30 s. Particle size distribution (Dv (10), Dv (50), (Dv (90)) was automatically calculated by Spraytec software version 3.1 (Malvern instrument, Malvern, UK, 2009). All determinations were performed in triplicate.

### 2.4. Residual Volumes, Nebulization Times, and Output Rates

To estimate residual volumes, nebulization times, and output rates, all nebulizers were operated in accordance with the manufacturer’s recommendations. Dornase alfa was kept on ice prior to nebulization. After filling the reservoir of each nebulizer with saline or dornase alfa, nebulizers were operated for the duration of nebulization: jet nebulizers were run continuously until 1 min after the beginning of sputtering, and mesh nebulizers were run until the end of the operating period. Nebulizer weights were measured before and after nebulization, and nebulization times were also recorded. Residual volumes were determined gravimetrically; evaporation was negligible [33]. Upon completion of nebulization, the output rate of each nebulizer was calculated using the equation below. This process was repeated three times for each nebulizer.
$$Output\;Rate\;\left(\frac{mL}{min}\right)=\frac{charged\;volume\;\left(mL\right)-residual\;volume\;\left(mL\right)}{duration\;of\;nebulization\;\left(min\right)}$$

### 2.5. Activity and Quantification of Dornase Alfa after Nebulization

To measure dornase alfa activity after nebulization, control (non-nebulized dornase alfa) samples (nominally 100% dornase alfa) were generated by transferring dornase alfa ampule contents at 2–8 °C into microtubes. Dornase alfa aerosols were collected using disposable plastic bags during nebulization and immediately transferred to clean microtubes and stored at 2–8 °C until required for analysis. The enzymatic activities of dornase alfa samples were determined using a kinetic fluorometric DNase activity assay. Briefly, collected dornase alfa samples were diluted 250-fold in saline, and then a 5 μL aliquot was transferred into the well of an analyzer containing 75 μL of nuclease-free water, 10 μL of 1X NucleaseAlter buffer, and 10 μL of DNase Alert Substrate solution. Fluorescence intensity was measured at Ex/Em = 535/556 nm at 1 min intervals for 1 h using a SpectraMax M3 microplate reader (Molecular Devices, Sunnyvale, CA, USA). Assays were performed at 37 °C. Relative DNase activity was calculated from the slopes of graphs of fluorescence intensity. The DNase activity of non-nebulized nominal dornase alfa was regarded as 100%, as shown in Table 4. The protein content of collected dornase alfa was measured with a DC protein assay kit using BSA as the standard. The absorbance values of the samples were measured after 15 min of incubation at RT at 750 nm using a SpectraMax M3 microplate reader.

### 2.6. Polyacrylamide Gel Electrophoresis (PAGE) of Dornase Alfa after Nebulization

Electrophoresis was carried out using 4–20% Mini-PROTEAN^®^ TGX™ Precast Gels (native polyacrylamide gels) or 15% sodium dodecyl-sulfate (SDS) polyacrylamide gels. Protein samples (10 μg/20 μL) for native and SDS polyacrylamide gel analyses were prepared using 5X native sample loading buffer and 5X SDS-PAGE sample loading buffer, respectively. The mixtures with 5X SDS-PAGE sample loading buffer were heat-treated for 5 min in a heating block before being loaded into SDS polyacrylamide gel. After separation by electrophoresis, gels were stained with Coomassie Brilliant Blue (CBB) solution containing 0.1% (w/v) CBB R-250, 50% (v/v) methanol, and 10% (v/v) acetic acid at RT overnight. Subsequently, the CBB-stained gels were placed in the destaining solution containing 40% (v/v) methanol and 10% (v/v) acetic acid several times. Images of gels were captured using a digital camera.

### 2.7. Temperature Measurement During Nebulization

The temperature changes of dornase alfa during nebulization were measured with a K-type thermocouple probe using a FLUKE-87-5 digital multimeter (Fluke, Everett, WA, USA). For the two jet nebulizers and the vibrating mesh nebulizer, a thermocouple probe was placed at a height of 2 mm from the center of the reservoir base, whereas for the static nebulizer, the probe was placed 2 mm from the horn. Dornase alfa was stored on ice prior to nebulization. Each nebulizer was operated after loading 2.5 mL of cold dornase alfa in the reservoir. Temperatures were recorded at 1 min intervals during nebulization. During measurements, the temperature and relative humidity were set at 23 °C and 43%, respectively.

### 2.8. Treatment of Mice with Dornase Alfa and Bronchoalveolar Lavage Fluid Collection

The mice were exposed to dornase alfa (2.5 mg/2.5 mL) in a closed inhalation exposure chamber (Figure 1) that was designed specifically for exposure of three mice at once. The chamber consisted of clear acrylic sheets (3 mm in thickness) with external dimensions of 20 × 20 × 15 cm, an internal volume of 6 L, and three small holes for head-only exposure [34]. Mice were randomly divided into five groups (*n* = 3/group), as follows: (i) untreated (negative control); (ii) nebulized by JN-PARIr; (iii) nebulized by JN-PARIb; (iv) nebulized by SMN-U150; (v) nebulized by VMN-SM1. Mice were anesthetized with intraperitoneal injections of ketamine (100 mg/kg) and xylazine (10 mg/kg) prior to undergoing nebulization. The delivery efficiency of each of the four nebulizers was examined under two different exposure conditions: (1) 2.5 mL of dornase alfa was placed in the reservoir and then nebulizers were operated for the duration of nebulization (until the onset of sputter or to dryness) or (2) to maintain a constant output, 2.5 mL or 3 mL of dornase alfa was charged into the jet nebulizer reservoir or mesh nebulizer reservoir, respectively, and then the nebulizers were run until the output volume reached 1.7 mL. Following dornase alfa exposure, bronchoalveolar lavage fluid (BALF) samples were harvested to assess the dornase alfa concentration in the lungs. The BALF samples were collected from each animal by cannulation of the exposed trachea and gentle flushing of the lungs with 0.7 mL of warm saline. BALF samples were centrifuged and supernatants were transferred to pre-cooled tubes and immediately frozen at −70 °C until required for analysis.

### 2.9. Determination of Dornase Alfa Levels in BALF

Dornase alfa levels in BALF samples were examined by conventional Western blot analysis using anti-DNase I. Briefly, BALF samples and 10 ng of dornase alfa proteins (i.e., positive control) were loaded into SDS polyacrylamide gel and then gel electrophoresis was performed. After transfer to polyvinylidene difluoride membranes, dornase alfa proteins were detected with primary and HRP conjugated secondary antibodies and Immobilon Western detection reagents, as described previously [35]. Chemiluminescence images were obtained and analyzed with ChemiDoc XRS + system and Image Lab software (Bio-Rad Laboratories, Hercules, CA, USA), respectively.

### 2.10. Statistical Analysis

One-way analysis of variance followed by Dunnett’s test was used to determine the significance of intergroup differences. Statistical analysis was performed using SigmaPlot software version 13 (Systat Software, San Jose, CA, USA, 2014). *p*-values of <0.05 were considered to be statistically significant.

## 3. Results

### 3.1. Nebulizer Characterization and Performance

To investigate the functional performance levels of jet and mesh nebulizers, we compared the mass median diameter (MMD) of aerosol droplets of saline or 2.5 mg/2.5 mL dornase alfa solution. The data showed that aerosol droplet sizes differed between nebulizers (Table 2). However, the Dv (50) results of dornase alfa were similar or slightly greater than those of saline. The Dv (50) results of dornase alfa generated by JN-PARIr and JN-PARIb clearly showed a difference between nozzle types: 3.18 ± 0.08 µm for the red nozzle and 4.99 ± 0.08 µm for the blue nozzle. The size difference according to the nozzle is similar to the results shown by the manufacturer. The Dv (50) results of dornase alfa for SMN-U150 and VMN-SM1 were 7.23 ± 0.07 µm and 5.93 ± 0.23 µm, respectively. Thus, JN-PARIr generated the smallest droplets and SMN-U150 generated the largest.

We examined the residual volumes, nebulization times, and output rates after filling reservoirs with saline or dornase alfa. Table 3 shows that the results for dornase alfa were similar to those for saline. Residual volumes for dornase alfa were approximately 24% for JN-PARIr and 27% for JN-PARIb, respectively, whereas the residual volume for VMN-SM1 was small enough to be negligible (Table 3). Incomplete aerosolization of the drug solution in the reservoirs of jet nebulizers is already a well-known characteristic [12,13,14]. The residual volume of SMN-U150 was found to be 15% higher than that of VMN-SM1, due to the protruding structure around the horn. The nebulization times of SMN-U150 for saline and dornase alfa were the shortest at 6.22 ± 0.076 and 6.47 ± 0.031 min, respectively, whereas JN-PARIr had the longest nebulization times (13.19 ± 0.072 min for saline and 13.30 ± 0.068 min for dornase alfa). The data showed that nebulization times varied between nebulizers. As a result, the nebulization time of JN-PARIr was 2-fold longer than that of SMN-U150. The output rates of dornase alfa for the JN-PARIr, JN-PARIb, SMN-U150, and VMN-SM1 nebulizers were 0.140 ± 0.002, 0.170 ± 0.004, 0.311 ± 0.002, and 0.238 ± 0.002 mL/min, respectively: the same order as that observed for aerosol droplet sizes (Table 3). These results show that the mesh nebulizers had smaller residual volumes and shorter nebulization times than the jet nebulizers.

### 3.2. Assessment of Dornase Alfa Activity, Concentration, and Structural Integrity after Nebulization

The enzyme activity of the collected dornase alfa aerosol from the four nebulizers was fully retained when compared with the non-nebulized nominal samples (Table 4). Likewise, the protein concentrations in the collected samples and in non-nebulized nominal samples were similar.

Finally, to investigate the structural integrity of dornase alfa in the collected aerosol samples, we performed two different gel electrophoresis tests. As demonstrated in Figure 2b, the SDS-PAGE results showed that the non-nebulized nominal sample had one major band at ~35 kDa, which concurs with previous SDS-PAGE and Western blot studies on dornase alfa [1,15]. However, native PAGE produced one major band larger than 35 kDa and a smear band between 40 and 60 kDa for the non-nebulized control, probably because dornase alfa is a glycoprotein and native PAGE was carried out without SDS (Figure 2a). Nevertheless, aggregated or degraded protein bands of dornase alfa were not observed under non-reducing or reducing conditions, which showed that the nebulization processes of the four nebulizers did not result in any aggregation or degradation of dornase alfa (Figure 2a,b). This in vitro analysis demonstrated that static and vibrating mesh nebulizers are capable of stably producing fully active dornase alfa aerosols in vitro without aggregation or degradation, similar to the jet nebulizers.

### 3.3. Investigation of Temperature Change in the Reservoir During the Operation of the Nebulizer

Temperature changes in the reservoir during nebulization were measured, as these can affect protein activity and structure [36,37]. Temperature changes in the reservoirs of the four nebulizers during nebulization are shown in Figure 3. The temperature of the dornase alfa solution in the JN-PARIr and JN-PARIb reservoirs quickly increased to 16 °C within 1 min, but remained at 16 °C below RT until nebulization was complete. The temperatures of SMN-U150 and VMN-SM1 continuously increased to 27 and 30 °C, respectively, higher than the RT. VMN-SM1 achieved a higher temperature because its nebulization time was greater than that of SMN-150, as shown in Table 3. Thus, the mesh nebulizers can significantly increase the temperature of dornase alfa solution in the reservoir to above RT, whereas jet nebulizers do not.

### 3.4. Delivery Efficiency of Dornase Alfa in Mice

Large molecules such as dornase alfa can be measured in BALF samples after exposure to dornase alfa, because they have little or no systemic absorption [36]. Mice were exposed to the dornase alfa under two different exposure conditions: Exposure 1 and 2. In the case of Exposure 1, nebulizer outputs from the four nebulizers were different, resulting in different residual volumes, as shown in Table 3. Thus, for Exposure 1, it is difficult to accurately compare the drug delivery efficiency because the residual volumes varied depending on the type of nebulizer. Exposure 2 was designed to make delivery efficiency easier to compare by setting the output to 1.7 mL for all four nebulizers. The output was set at 1.7 mL because the largest residual volume of JN-PARIb was measured at approximately 0.7 mL with 2.5 mL of dornase alfa (Table 3). As shown in Figure 4a,b, the JN-PARIr nebulizer achieved a significantly higher level of dornase alfa in BALF than the other nebulizers under both Exposure 1 (JN-PARIr: 552, JN-PARIb: 248, SMN-U150: 133, VMN-SM1: 360 ng/mL) and Exposure 2 (JN-PARIr: 549, JN-PARIb: 252, SMN-U150: 125, VMN-SM1: 259 ng/mL), while SMN-150, which had the highest output rate and the shortest nebulization time, had the lowest delivery efficiency under both exposure conditions. In Exposure 1, the level of dornase alfa in BALF for the VMN-SM1 nebulizer was generally higher than that of the JN-PARIb nebulizer, but for Exposure 2, VMN-SM1 and JN-PARIb had similar delivery efficiencies. Although JN-PARIr had a high residual volume and a long nebulization time, it showed the highest dornase alfa delivery efficiency under both conditions. Interestingly, the pattern of efficiency for the delivery of dornase alfa into lungs at Exposure 2 was similar to the order of particle size of dornase alfa aerosol shown in Table 2. These results indicate that the delivery efficiency of dornase alfa is more related to the droplet size than the nebulizer output.

## 4. Discussion

Dornase alfa is indicated for daily administration using a recommended nebulizer in conjunction with standard therapies for the management of CF patients to improve pulmonary function. Currently, product information on dornase alfa recommends only six jet nebulizer/compressor systems and one vibrating mesh nebulizer [15]. These nebulizers generate different droplet sizes and have different output characteristics and dosing rates, which largely depends on the mode of operation.

It has been well established that total and regional lung deposition of inhaled aerosols are determined by two key factors: droplet size and output [37,38]. In the present study, we investigated the in vitro performance and drug stability of jet and mesh nebulizers for dornase alfa, and then compared their delivery efficiency in a mouse model.

The droplet sizes produced by jet nebulizers were generally smaller than those generated by mesh nebulizers (Table 2). Interestingly, for jet nebulizers, the droplet sizes are directly proportional to the size of the nozzle [39]. We found that the sizes of droplets produced by the four nebulizers studied were greater than those specified by the manufacturers. This was due to the use of different laser diffraction instruments or the use of mass median aerodynamic diameters (MMAD), which were measured using an air-driven cascade impactor [32]. In fact, it has been shown that the MMDs of droplets as determined by laser diffraction measure larger than droplets measured by MMAD due to evaporation losses from small droplets at the plume edge [40]. Therefore, the droplet sizes may vary slightly according to the particular analytical method used. According to the performance data shown in Table 3, the residual volumes of JN-PARIr and JN-PARIb were significantly greater than those of mesh nebulizers, and this resulted in lower output rates. The nebulization time of JN-PARIr was the longest, while that of SMN-U150 was the shortest. Although the nebulization time of JN-PARIb was similar to that of VMN-SM1, its output rate was lower because JN-PARIb had a high residual volume. This result is consistent with previous studies that compared the performance of jet and mesh nebulizers [41,42].

Measurements of enzyme activity, protein concentration, and structural integrity of dornase alfa after nebulization are essential because they can impact clinical outcomes significantly. Also, aggregation or degradation of a protein could decrease its activity or even result in the generation of an immunogenic protein [43,44]. The enzymatic activity and protein concentrations of collected dornase alfa aerosols from the four nebulizers were similar to those of non-nebulized nominal samples (Table 4). In addition, the results of PAGE using native polyacrylamide gel and SDS polyacrylamide gel showed that no larger or smaller fragments of dornase alfa were observed after nebulization, which demonstrates that dornase alfa was nebulized effectively without aggregation or degradation by all four nebulizers. These results show that dornase alfa remains in its active form and is analytically unaltered following aerosolization in all four nebulizers.

Various nebulizers have been developed to deliver small-molecule drugs to the lungs rather than peptides or proteins. It is well known that jet nebulizers generally decrease the reservoir’s solution temperature during the first 2 min of operation, whereas ultrasonic nebulizers increase the temperature by ~20 °C within a few minutes [45]. Mesh nebulizers are based on further development of the ultrasonic nebulization principle to improve aerosol generation efficiency, speed of nebulization, and device handling, as well as to reduce residual volumes; however, optimizing protein stability during nebulization has not been not a primary concern [31,46]. As research began to deliver expensive proteins directly into the lungs using a nebulizer, researchers examined whether proteins were stably nebulized with various type of nebulizers, because aggregation and degradation of proteins can reduce protein activity or even result in immunogenicity [43,44].

Mesh nebulizers are generally considered to enable aerosol generation at low shear, and thus are suitable for use with biopharmaceuticals. Several reports using small-molecule drugs have reported that mesh nebulizers show little or no change in the temperature of the reservoir solution. Since these studies use drugs stored at RT, most of the temperature changes start from RT [47]. However, current reports issued to date on the heating of protein drugs are inconsistent. A recent study using cold storage proteins reported reservoir temperatures reaching more than 35 °C at the end of nebulization [31]. In the present study, we investigated the temperature changes of cold dornase alfa in the reservoir during the operation of the nebulizer (Figure 3), because dornase alfa, unlike small-molecule drugs, must be stored according to the manufacturer’s recommended temperature (range 2–8 °C) until the point of administration. The results of temperature changes using 2.5 mL of cold dornase alfa solution clearly showed that static and vibrating mesh nebulizers increase the temperature inside the reservoir above RT during nebulization (Figure 3). Since dornase alfa has a high melting temperature of 67.4 °C, its activity in aerosols produced by these nebulizers was maintained despite the reservoir temperature increasing to 30 °C [48]. This result shows that the longer the nebulization time, the greater the operating energy of the nebulizer, which can cause the reservoir temperature to rise significantly. Therefore, heat-sensitive proteins in particular should be considered for use alone or in combination with other drugs.

Accurate knowledge of delivery efficiency to the lungs after drug administration by nebulizer is critical for establishing dose–response correlations and optimizing the treatment outcomes [49]. The amount of dornase alfa delivered by nebulizers has usually been determined by capturing aerosols with an absolute filter connected to an in vitro breathing simulator developed to simulate in vivo breathing [20,21]. However, this method has shortcomings in terms of determining the delivery efficiency to the lungs, as it involves collecting all of the aerosol generated, and thus does not accurately mimic mouth-to-lung delivery. Aerosol droplet size is one of the most important variables in terms of determining the dose deposited and the distribution of the drug in the lungs. Most therapeutic aerosols generate a wide range of droplet sizes. It has been well established that fine droplets of less than 5 µm are deposited within the peripheral airways, whereas larger droplets typically impact the oropharynx and are eventually swallowed [50]. In addition, clinical studies on dornase alfa have only evaluated pulmonary functions and did not define the aerosol dose of dornase alfa delivered to the lungs. Therefore, there is a need to evaluate the delivery efficiency of dornase alfa reaching the lungs by nebulizer using an in vivo model. In this study, we decided to use an in vivo mouse model to compare the lung delivery efficiency of nebulizers that utilize different operating principles, generate different aerosol droplet sizes, and have different residual volumes and output rates.

Dornase alfa levels in the BALF samples obtained from mouse lungs under two different exposure conditions were not detectable using commercial hDNase I ELISA kits due to a mismatch between the recombinant protein and the coated antibody or detection antibody. In addition, therapeutic proteins generally have high molecular weights and cannot easily penetrate the capillaries in the lungs. Thus, they can be detected in BALF [36]. When examining a nebulizer’s drug delivery efficiency under in vitro or in vivo conditions, nebulizers are generally operated for the duration of nebulization. However, these conditions make it difficult to compare drug delivery efficiencies, because the output rate and nebulization time depend on the nebulizer type. In this study, to compare the delivery efficiency of dornase alfa, mice were exposed to dornase alfa under Exposure 1 (duration of nebulization) and Exposure 2 (constant output of 1.7 mL) conditions. Under both exposure conditions, the JN-PARIr nebulizer produced the highest level of dornase alfa in BALF (Figure 4a,b). On the other hand, the SMN-U150 nebulizer, which generated the largest aerosol droplets and had a lower residual volume and the highest output rate, indicated the lowest level of dornase alfa under both exposure conditions. The in vivo study of Exposure 2 (constant output of 1.7 mL), shown in Figure 4b, indicated that the efficiency of delivering dornase alfa to the lungs was overwhelmingly determined by the aerosol droplet size, which concurs with the findings of previous studies that used a higher lung deposition of aerosols with smaller particle sizes [11].

Although in vivo data are more relevant than in vitro data, our mouse model also has its limitations. Anesthetization may have caused a significant departure from clinical conditions, and airway branching in the human lung is relatively symmetric, whereas that of mouse lungs is distinctly asymmetric or monopodial [50]. Despite these differences, mouse lungs are being increasingly used as surrogates of human lungs in studies on the deposition and risks posed by inhaled particles. Nevertheless, we caution that the limitations of the anesthetized mouse model should be considered when interpreting our data.

## 5. Conclusions

Our results showed no overall loss of enzymatic activity or protein content and no increase in aggregation or degradation following aerosolization by the four studied nebulizers. We also found the jet nebulizers had lower output rates than the mesh nebulizer, but produced smaller aerosol droplets, which resulted in higher efficiency of dornase alfa delivery to the lungs. Static and mesh nebulizers were found to increase the reservoir temperature to close to 30 °C when nebulized with 2.5 mL of dornase alfa. Therefore, when using a mesh nebulizer to deliver protein biopharmaceuticals into the lungs, manufacturers should fully consider the prescribed medication volume and melting temperature. We suggest clinical and analytical studies be undertaken to confirm the compatibilities of medications and nebulizers.

## Figures and Tables

**Figure 1 pharmaceutics-12-00721-f001:**
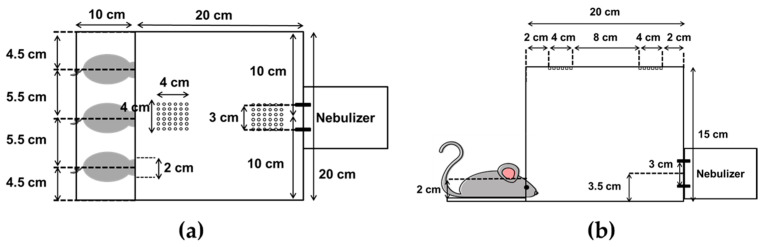
A head-only exposure chamber system for mice: (**a**) top view and (**b**) side view.

**Figure 2 pharmaceutics-12-00721-f002:**
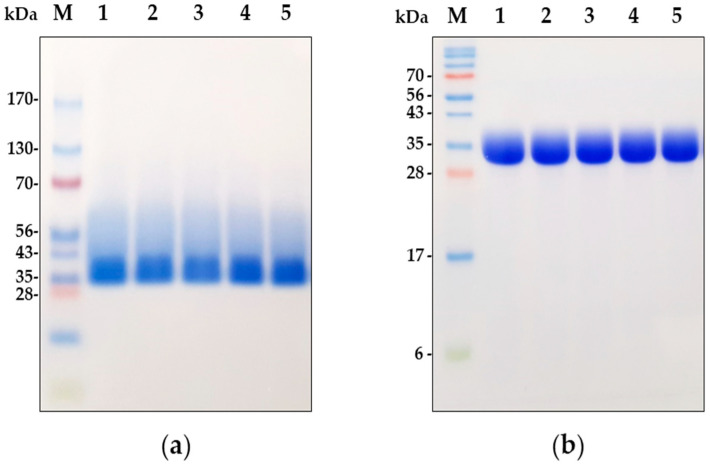
Representative native PAGE (**a**) and SDS-PAGE (**b**) analyses of dornase alfa aerosols generated by the four nebulizers: molecular weight standard (**lane M**), non-nebulized nominal (**lane 1**), JN-PARIr (**lane 2**), JN-PARIb (**lane 3**), SMN-U150 (**lane 4**), and VMN-SM1 (**lane 5**).

**Figure 3 pharmaceutics-12-00721-f003:**
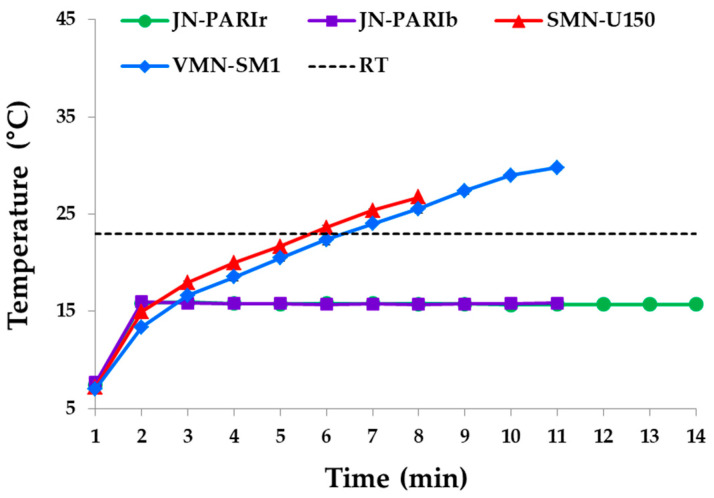
Plot of reservoir temperatures during nebulization by the JN-PARIr (**green circle line**), JN-PARIb (**purple square line**), SMN-U150 (**red triangle line**), and VMN-SM1 (**blue diamond line**) nebulizers. The black dashed lined indicates room temperature (RT).

**Figure 4 pharmaceutics-12-00721-f004:**
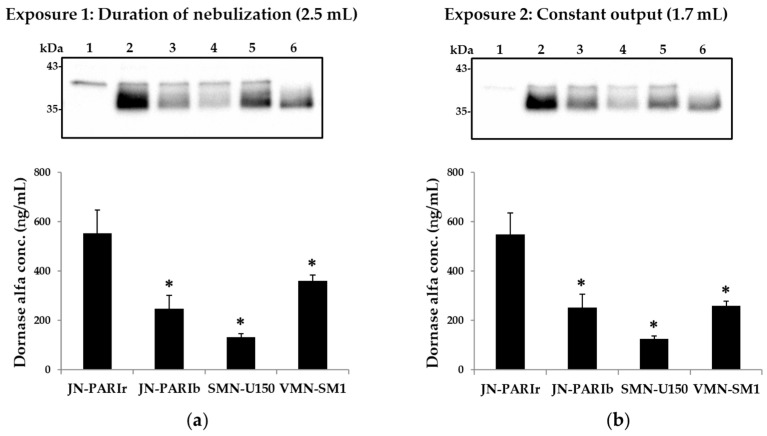
Representative Western blot images and levels of dornase alfa in bronchoalveolar lavage fluid (BALF) from mice treated using the four nebulizers. (**a**) Duration of nebulization with 2.5 mL of dornase alfa (Exposure 1); (**b**) Constant output at 1.7 mL (Exposure 2). Normal mice BALF (**lane 1**), JN-PARIr-treated mice (**lane 2**), JN-PARIb-treated mice (**lane 3**), SMN-U150-treated mice (**lane 4**), and VMN-SM1-treated mice (**lane 5**). Lane 6 shows the blot obtained for 10 ng of dornase alfa (**lane 6**). All values are means ± SD. * *p* < 0.05 versus JN-PARIr.

**Table 1 pharmaceutics-12-00721-t001:** Nebulizers used in the study.

Mode of Operation	Model	Abbreviation in the Study
Jet	PARI BOY SX-red nozzle (PARI GmbH, Starnberg, Germany)	JN-PARIr
	PARI BOY SX-blue nozzle (PARI GmbH, Starnberg, Germany)	JN-PARIb
Static mesh	NE-U150 (Omron Healthcare, Kyoto, Japan)	SMN-U150
Vibrating mesh	NE-SM1 NEPLUS (KTMED Co., Seoul, Korea)	VMN-SM1

**Table 2 pharmaceutics-12-00721-t002:** Aerosol droplet sizes for saline and dornase alfa generated by the four nebulizers.

Device	Saline				Dornase Alfa			
	Dv10 (µm)	Dv50 (µm)	Dv90 (µm)	Span	Dv10 (µm)	Dv50 (µm)	Dv90 (µm)	Span
JN-PARIr	1.64 ± 0.03	3.05 ± 0.08	6.30 ± 0.26	1.83 ± 0.01	1.55 ± 0.05	3.18 ± 0.08	7.08 ± 0.32	1.81 ± 0.01
JN-PARIb	1.89 ± 0.01	4.48 ± 0.01	10.18 ± 0.18	1.86 ± 0.03	1.85 ± 0.02	4.99 ± 0.08	12.61 ± 0.29	1.93 ± 0.02
SMN-U150	2.77 ± 0.12	6.69 ± 0.09	16.74 ± 0.73	1.97 ± 0.10	3.02 ± 0.02	7.23 ± 0.07	16.69 ± 0.39	1.99 ± 0.03
VMN-SM1	2.17 ± 0.04	5.18 ± 0.18	12.60 ± 0.17	1.98 ± 0.07	2.39 ± 0.05	5.93 ± 0.23	13.64 ± 0.14	2.00 ± 0.08

**Table 3 pharmaceutics-12-00721-t003:** Residual volumes, nebulization times, and output rates for the four nebulizers.

Device	Residual Volume (mL)		Nebulization Time (min)		Output Rate (mL/min)	
	Saline	Dornase Alfa	Saline	Dornase Alfa	Saline	Dornase Alfa
JN-PARIr	0.595 ± 0.014	0.609 ± 0.024	13.19 ± 0.072	13.30 ± 0.068	0.143 ± 0.002	0.140 ± 0.002
JN-PARIb	0.639 ± 0.015	0.687 ± 0.027	10.19 ± 0.066	10.32 ± 0.183	0.181 ± 0.002	0.170 ± 0.004
SMN-U150	0.374 ± 0.011	0.385 ± 0.007	6.22 ± 0.076	6.47 ± 0.031	0.334 ± 0.008	0.311 ± 0.002
VMN-SM1	0.034 ± 0.005	0.047 ± 0.002	10.09 ± 0.036	10.15 ± 0.057	0.243 ± 0.001	0.238 ± 0.002

**Table 4 pharmaceutics-12-00721-t004:** Dornase alfa activity and concentrations of nebulized samples expressed as percentages of non-nebulized nominals.

Device	Enzyme Activity (%)	Protein Concentration (mg/mL)
Nominal	100	1.006 ± 0.001
JN-PARIr	99.53 ± 2.88	1.000 ± 0.011
JN-PARIb	98.63 ± 0.63	0.995 ± 0.025
SMN-U150	100.37 ± 1.32	0.995 ± 0.008
VMN-SM1	100.34 ± 1.12	0.995 ± 0.005

There were no statistically significant differences in the enzyme activity and protein concentration between the nominal and experimental groups (*p* < 0.05).

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
