# Peer review of "Aerosol Delivery of Dornase Alfa Generated by Jet and Mesh Nebulizers"

_pharmaceutics, 2020, doi:10.3390/pharmaceutics12080721_

Round 1

Reviewer 1 Report

This study compared four nebulizers on their ability to nebulize and deliver the commercially available formulation of dornase alfa (Pulmozyme), including an in vivo evaluation of delivered dose to lungs of mice. While the key finding on the biological integrity of nebulized  protein was encouraging, certain important aspects of analysis might be lacking. Kindly find the following comments and suggestions for your consideration:

Major comments:

  1. The inclusion of illustrative figures on the exposure chamber used in animal models [l. 181 - 182], and preferably on the design of the two types of mesh nebulizers [l. 71 - 72], should improve overall presentation on the experimental setup. In fact, the design of the exposure chamber could significantly contribute to the observed results of in vivo drug deposition, and detailed description or validation should be provided.
  2. In addition to Dv50, Dv10, Dv90 and the span value should at least be presented to describe the size distribution of the nebulized aerosols. 
  3. The aerosol particle size distribution of the various nebulizers should be properly measured using suitable cascade impactors, complemented by the size distribution results as measured using laser diffraction. 
  4. The overall values of the study would be much enriched and the context of the study would be better put in if the experiments were also conducted on the recommended/approved devices (preferably one nebulizer/compressor combination and the nebulizer system) and the results compared. 

Minor comments:

  1. Significant figures (of numbers) should be reported consistently (including those in the main text and tables, etc) and as much as relevant/needed. 
  2. [Section 3.2] Any statistical tests performed to support the claim of "full retained" enzyme activity or "similar" protein concentrations?
  3. [l.233] Minor typographical mistakes in references?
  4. What was the recovery in the collection of BALF? How consistent it was? Could the differences in dornase alfa concentration in BALF be due to the variations in BALF collection?
  5. The article was mostly clearly written, yet further proofreading is required (e.g. l.338 - 339, inconsistent italicisation on in vitro or in vivo, etc.).

Author Response

We greatly appreciate the reviewer’s kind and insightful comments on our manuscript (Manuscript ID: pharmaceutics 840313). Accordingly, the manuscript has been revised and clarified. The manuscript has undergone English language editing by MDPI (ID 20164). The text has been checked for correct use of grammar and common technical terms, and edited.

Please find attached response.

Reviewer 2 Report

The manuscript deals with the comparison of aerosol delivery of dornase alfa from different nebulizers, evidencing different behavior. 

The influence of nebulizer's type on the aerosol delivery has been reported also in other papers that are not reported here (i.e. Adorni et al., Pharmaceutics 2019, 11, 406).

However, I suggest an accurately revision of English style.

Moreover,

  1. line 122: model of Spraytec used
  2. line 125: please describe in detail how the aerosol droplet size determination was performed
  3. line 134: what means "onset of sputtering"? The sputtering can be seen only with jet nebulizers, not with mesh nebulizers. Moreover, in the case jet nebulizer, according to guidelines, the nebulization is stopped 1 minute after the sputtering.
  4. the output rate have to be measured after 1 min of nebulization process.
  5. line 146: how the dornase alfa aerosol was collected in the disposable plastic bags? How the aerosol was transferred in the microbes?
  6. line 228: in the case of jet nebulizers, the nebulization has to be stopped 1 minute after sputtering as the size of the droplet change.
  7. line 232: what do you mean "well-known as a characteristic"?
  8. lines 274-275: what are "The static and vibrating mesh nebulizers"?
  9. Figure 2: no error bars are reported. Does it mean that a single analysis has been carried out?
  10. section 3.4: some information, i.e exposure conditions are part of the method description, then they have to be reported in the material's section.
  11. I did not understand the purpose of the Exposure 1 and Exposure 2, also in Figure 3.
  12. line 349-351: the reported droplet size depends on the method used, then can be compared with your data is the method is different.
  13. line 449: what do you mean with "melting temperature of a solution
  14. the conclusions are missing.

Author Response

(The authors gave the same response as above.)

Round 2

Reviewer 1 Report

The comments have been responded accordingly. The manuscript can be improved by including suitable responses into the main text, for example, by mentioning the unavailability of the recommended devices in Korea, it could help better define and set the ground for the work that otherwise lacks significant scientific novelty.

I have no further comments on the work. 

Author Response

We greatly appreciate the reviewer’s kind and insightful comments on our manuscript (Manuscript ID: pharmaceutics 840313). Accordingly, the manuscript has been revised and clarified.

- We added following sentences for reviewer 1’s comments in Materials and Methods (line: 103-106).

“Since the recommended devices for dornase alfa were not approved in Korea market, popular items were selected for this study among the nebulizers commonly used in clinical. PARI BOY SX in this study is the closest model with PARI LC PLUS which is one of the recommended devices for dornase alfa.”

Reviewer 2 Report

The authors revised the manuscript accordingly to the Reviewer's comment.

Author Response

We greatly appreciate the reviewer’s kind and insightful comments on our manuscript (Manuscript ID: pharmaceutics 840313). There is no further revision according to reviewer 2's comment.